# Interplay between hydrophilicity and surface barriers on water transport in zeolite membranes

Matteo Fasano[1], Thomas Humplik[2], Alessio Bevilacqua[1], Michael Tsapatsis[3], Eliodoro Chiavazzo[1], Evelyn N. Wang[2] & Pietro Asinari[1]

A comprehensive understanding of molecular transport within nanoporous materials remains elusive in a broad variety of engineering and biomedical applications. Here, experiments and atomistic simulations are synergically used to elucidate the non-trivial interplay between nanopore hydrophilicity and surface barriers on the overall water transport through zeolite crystals. At these nanometre-length scales, these results highlight the dominating effect of surface imperfections with reduced permeability on the overall water transport. A simple diffusion resistance model is shown to be sufficient to capture the effects of both intracrystalline and surface diffusion resistances, thus properly linking simulation to experimental evidence. This work suggests that future experimental work should focus on eliminating/overcoming these surface imperfections, which promise an order of magnitude improvement in permeability.

[1] Department of Energy, Politecnico di Torino, Corso Duca degli Abruzzi 24, Torino 10129, Italy. [2] Department of Mechanical Engineering, Massachusetts Institute of Technology, Cambridge, Massachusetts 02139, USA. [3] Department of Chemical Engineering and Materials Science, University of Minnesota, Minneapolis, Minnesota 55455, USA. Correspondence and requests for materials should be addressed to E.N.W. (email: enwang@mit.edu) or to P.A. (email: pietro.asinari@polito.it).

The large surface area to volume ratio, the unique nanoscale properties and the recent progress in synthesis of nanoporous solids have motivated the growing interest in using nanoporous materials in many applications, both in engineering and biomedical fields[1–6]. These characteristics can be beneficial to enhance the performance of desalination, energy storage devices, catalysts, molecular sieves, sensors and opto-electronic devices as well[7–14]. The common theme among these various applications is the relationship between mass transport properties and performance, which has stimulated fundamental research in understanding transport mechanisms in nanoporous materials.

For water purification membranes that use nano and microporous materials, two transport processes, namely the intracrystalline diffusion of guest molecules and the permeation through the surface, govern the rates of water uptake and release[3]. Although the peculiar mass transport properties of water confined within nanometric cavities (that is, water confined to pore diameters of $\leq 2\,nm$) have been extensively investigated by simulations and experiments[15–18], the quantitative analysis of the finite rate of permeation through the external boundaries of nanoporous particles has been only elucidated by recent advances in chemical imaging[19]. On the one side, traditional 'macroscopic' methods based on the water uptake/release given by a step pressure change allow estimating the overall (transport) diffusivity of guest molecules; on the other side, microimaging techniques are now paving the way towards understanding surface permeation mechanisms by directly measuring diffusion paths with molecular precision[19,20].

By measuring guest molecule diffusion within individual crystallites (microimaging techniques) rather than crystallite assemblages ('macroscopic' methods), the intracrystalline diffusion resistance was found to be only a part of a more complex series of transport resistances, including barriers in the nanoporous particle and on its surface[19,21]. In general, the transport diffusivities measured by 'macroscopic' methods are orders of magnitude lower than predictions from molecular simulations[22]. Microimaging techniques are now proving that the transport of guest molecules is also hindered by the 'non-ideality' of crystal structures, which show unusual surface pore restrictions (or total pore blockage) and inner crystal grain boundaries, according to the synthesis, storage and pretreatment of the considered samples[23,24].

Among the many nano- and microporous materials currently investigated, zeolites are attracting increasing attention, due to the variety of networks and sizes of the nanopores and the possibility of introducing defects or compounds to tune their heat and mass transfer properties[25]. Zeolites are characterized by a distribution of tight pores, which are typically smaller than the diameter of hydrated salt ions. Hence, zeolite-based membranes could completely reject salt ions, while allowing the permeation of water molecules, as for example in case of reverse osmosis (RO) processes[26,27]. However, a deeper investigation of phenomena underpinning water transport in sub-nanometre pores is desirable, to allow better design of zeolite-based membranes, and it may unlock their industrial potential in the exponentially growing desalination market[2,28].

The water diffusion in nanoporous membranes is ruled by physical (that is, volumetric capacity, pressure and temperature), geometric (that is, network and pore diameter) and chemical (that is, interactions between liquid and solid phase) properties[29]. In detail, the interaction between pore surface and water transport can be ruled by functional (for example, hydrophobic or hydrophilic) defects within the porous framework and thus mass transport coefficients of inner water can be precisely tuned. As permeability through a nanoporous membrane is affected by both equilibrium (solubility) and non-equilibrium (diffusivity) quantities, the property controlling the effective mass transport is still debated[27]. Although it is well-established that hydrophilicity in pores increases water sorption[27,30], some recent studies have also shown a simultaneous decrease in the diffusion coefficient for the intruded water[27]. However, evidence that hydrophobic nanoporous membranes demonstrate enhanced permeability compared with hydrophilic membranes[27] is under debate in literature, where opposite trends have been observed[26,31]. This discrepancy indicates that experiments alone may be insufficient to obtain a comprehensive understanding of the physics of nanoconfined water. In fact, even small differences in the local composition, crystal size or active layer thickness of the considered membrane can significantly alter the measured quantities. Furthermore, lesser-controlled factors such as contamination, intercrystalline mesoporosity, surface barriers (for example, pore blockages or narrowing) can also substantially affect the measured results. Although the rapid development of advanced physical characterization tools will soon allow a more thorough examination of the properties of nanoconfined water[32], the current capabilities of molecular dynamics (MD) simulations are allowing the investigation of these nanoscale transport phenomena to provide experimental guidelines for tuning the membrane permeability[33–35].

In this study, water transport in nanoporous materials is investigated through experiments and MD simulations. In particular, we model the role of hydrophilic defects within the hydrophobic framework of pristine mordenite framework inverted (MFI) zeolites, namely silicalite-1, on the intracrystalline mass transfer of intruded water by means of zeolite–water interactions. Although hydrophilic defects lead to a significant reduction of intracrystalline water diffusion at low pore fillings (that is, adsorption pressures), water transport is not affected by framework hydrophilicity at larger hydrations. Furthermore, we find that modelling and experimental diffusivities differ by several orders of magnitudes, which is hypothesized to be a result of additional transport resistances (that is, surface diffusion resistances). The role of such surface barriers on the transport of water throughout zeolites of varying size is then experimentally investigated and physically interpreted via a diffusion resistance model, which is able to quantify the influence of both surface and volumetric diffusion resistances. The results indicate that these surface barriers induce an extreme rate-limiting transport resistance, being proportional to the concentration of defects within MFI zeolites. The diffusion resistance model suggests that even small reductions in these surface diffusion resistances can lead to orders of magnitude increases in water flux through membranes for engineering (for example, desalination) or biomedical (for example, molecular sieving or detection) applications.

## Results

**Mass transport at the nanoscale**. Permeability ($P$) is a measure of the fluid flux through a membrane with a given geometry and pressure gradient, and is given by the product of membrane solubility ($S$) and corrected diffusivity ($D_0$) of the fluid through the membrane, $P = SD_0$ (ref. 36). Solubility is defined as the variation of adsorbed ($p < p_0$) or infiltrated ($p > p_0$) water molecules ($\omega$) in the porous material due to pressure ($p$) changes, $S = \partial\omega/\partial p$, and can be obtained from adsorption/infiltration isotherms[27]. At the molecular scale, molecules are subjected to Brownian dynamics[37]. The self-diffusion coefficient ($D$) of a molecule characterizes its molecular mobility under equilibrium conditions and reflects the combined effect of molecule–wall and molecule–molecule collisions/interactions[38–40]. According to

Einstein's theory, $D$ can be evaluated from the mean square displacement of single molecules[41]. In the Fick's law, the diffusive flux of fluid is related to a concentration gradient at steady state, $j_x = -D_T \frac{\partial c}{\partial x}$, where $D_T$ is the Fick's diffusion coefficient, $j_x$ and $c$ are the diffusion flux and concentration of the fluid along the direction $x$, respectively. Even though chemical potential gradient is the fundamental driving force for diffusion, concentration gradient is the driving force for the Fickian diffusion model. However, it can be easily demonstrated that the Fickian diffusion coefficient ($D_T$, defined with the concentration gradient) and thermodynamically corrected diffusivity ($D_0$, defined with the chemical potential gradient) are related by $\Gamma$, the thermodynamic factor:

$$D_T = \Gamma D_0, \tag{1}$$

where $\Gamma = \partial \ln p / \partial \ln \omega$ is the gradient in logarithmic coordinates of the sorption isotherm[19,42]. From a molecular standpoint, $D_0$ quantifies the collective motion of molecules and it serves as the contact point between $D_T$ (measured by experiments) and Maxwell–Stefan equations (molecular diffusion from first principles)[40]. The Maxwell–Stefan equations demonstrate that the self-diffusion coefficient of a fluid confined in a pore is mainly limited by resistances due to solid–fluid interactions ($1/D_0$) and fluid–fluid interactions ($1/Đ_{ii}$):

$$1/D = 1/D_0 + 1/Đ_{ii}, \tag{2}$$

where $Đ_{ii}$ the self-exchange coefficient[39].

**Intracrystalline diffusion in zeolites**. It has been shown that the insertion of hydrophilic defects (via surface coatings or through synthesis procedures) within a hydrophobic nanopore significantly affects the mass transport of intruded water molecules[43]. In our previous work, controlled sorption and infiltration experiments showed that the diffusivity and permeability of water decreased by over an order of magnitude through the insertion of hydrophilic defects into a previously pristine MFI zeolite structure[27]. For deeper mechanistic understanding of this reduction in diffusivity, here we investigated the self-diffusivity of water nanoconfined in defected MFI crystals at several pore hydrations by MD simulations.

Pristine MFI zeolite (silicalite-1) crystals are made of only silicon and oxygen atoms (Fig. 1a,b)[44]. MFI zeolites with increasing hydrophilicity are simulated by progressively introducing defects in the pristine structure of silicalite-1. For lab-synthesized MFI zeolites (Fig. 1c), the insertion of aluminium atoms creates hydrophilic point defects[45,46]. The replacement of silicon by aluminium promotes the presence of an unbound oxygen, which gives rise to a silanol terminal. The 'silanol nest model' proposed by Caillez et al.[30,47] can be then adopted as a first approximation of zeolite's defects with tunable hydrophilicity[47]. Hence, the increasing zeolite hydrophilicity due to Al insertion is mimicked by introducing silanols in the MFI pores (Fig. 1d).

The influence of defect concentration and silanol hydrophilicity on $D$ was then evaluated. MFI crystals with 0, 0.33, 0.89 and 3.06% substitutions of silicon atoms by aluminium ones and 'weak'[30] silanol nests ($q_H = 0.45e$ and $q_O = -0.9e$, where $e$ is the elementary charge) were studied. Moreover, an MFI zeolite with 'strong' silanol nests ($q_H = 0.65e$ and $q_O = -1.1e$) and 0.89% Al/Si substitutions was also investigated[30]. $D$ were then measured at the various pore hydration states. As shown in Fig. 2a, the measured $D$ was generally reduced as compared with bulk value ($D_B = 3.54 \times 10^{-9}$ m$^2$ s$^{-1}$, see Supplementary Note 1 for further details), although this was an expected result for water confined to nanometre-sized geometries[15]. From Fig. 2a, two

clear regimes of mass transport were evident and are a function of the hydration of the pores ($\vartheta_M = \omega/\omega_M$, where $\omega_M = 52$ N/UC is the maximum capacity of the framework in MFI pores, see Supplementary Figs 1–3). Results show that $D$ is sensible to both defect concentration and silanol hydrophilicity at low hydration values (that is, $\vartheta_M < 0.5$). The diffusivity decreased as the crystal hydrophilicity increased and the trend agreed with previous experiments[27,48–50]. However, at higher values of the pore hydration (that is, $\vartheta_M > 0.5$), $D$ becomes independent of the defect density. Furthermore, irrespective of the defect density, a progressive reduction in $D$ with increasing $\vartheta_M$ was observed[51,52]. A mechanistic interpretation of the above results is needed.

On average, the amount of water–water hydrogen bonds (H-bonds) per fluid molecule was obtained at different $\vartheta_M$ and defect concentrations (Supplementary Fig. 4). Although larger pore hydrations led to increased H-bonds and thus increased intracrystalline diffusion resistance, the constant number of H-bonds for $\vartheta_M \geq 0.5$ with several concentrations of defects confirmed the negligible influence of the surface hydrophilicity on $D$ at large pore fillings. Moreover, we investigated the specific interaction energies (see Supplementary Note 2) between water and MFI crystals given by Coulomb ($U_C$) or Lennard–Jones ($U_{LJ}$) interactions (Supplementary Fig. 5). As shown in Supplementary Fig. 6, $E_{wz}$ (water–zeolite-specific interaction energy) had a significant effect on water dynamics at low hydration regimes. Furthermore, higher $E_{wz}$ were measured within more hydrophilic structures, mainly because of stronger Coulomb interactions. The magnitude of $E_{wz}$ with respect to $E_{ww}$ (water–water-specific interaction energy) decreases following a power law, being $E_{ww}$ eventually predominant for $\vartheta_M > 0.5$. In addition, the radial distribution function of the solvent ($g(r)$) with respect to the surface of the pores was computed. Results in Supplementary Fig. 7a further confirm a significant influence of defects only at low pore hydration ($\vartheta_M = 0.10$), where $g(r)$ for the more hydrophilic zeolite demonstrated a peak at $r \cong 0.18$ nm due to silanol–water hydrogen bonds. This peak is a clear sign of solvent adsorption on the hydrophilic pore surface, which is responsible of reduced water mobility and thus decreased self-diffusivity. At large pore hydration ($\vartheta_M = 0.95$), instead, similar $g(r)$ for the pristine and defected MFI are observed (Supplementary Fig. 7b). On the one side, the self-diffusion coefficient of water in defected MFI is predominantly determined by water–zeolite interactions ($E_{wz}$) at low hydrations ($E_{ww}$). On the other side, $D$ is not influenced by pore characteristics at high hydration regimes, being mainly governed by water–water interactions (Fig. 2b). At low pore fillings, the interactions between solvent molecules are reduced respect to solid–liquid ones and thus equation (2) simplifies to $D \cong D_0$. This result explains the larger self-diffusivity of water measured in the more hydrophobic MFI frameworks, because $D_0$ is mainly dictated by solid–liquid interactions. Conversely, liquid–liquid interactions become predominant at large pore fillings; hence, $D \sim Đ_{ii}$ and the self-diffusion coefficient becomes dependent on the surface characteristics of pores. In summary, at the typical working pressures of current RO systems (that is, $P < 8$ MPa), the intracrystalline water diffusion in MFI is significantly affected by framework hydrophilicity; therefore, it can be tuned by Al defects introduction.

**Surface effects on diffusion**. Although trends of the self-diffusion coefficient (MD simulations) and the corrected water diffusivity (experiments) in MFI with defects agree in terms of a decreasing diffusivity with increasing hydrophilicity[27], the difference between the two values can show orders of magnitude discrepancies. However, the experimentally measured diffusivity of water is also strongly correlated to the distance over which

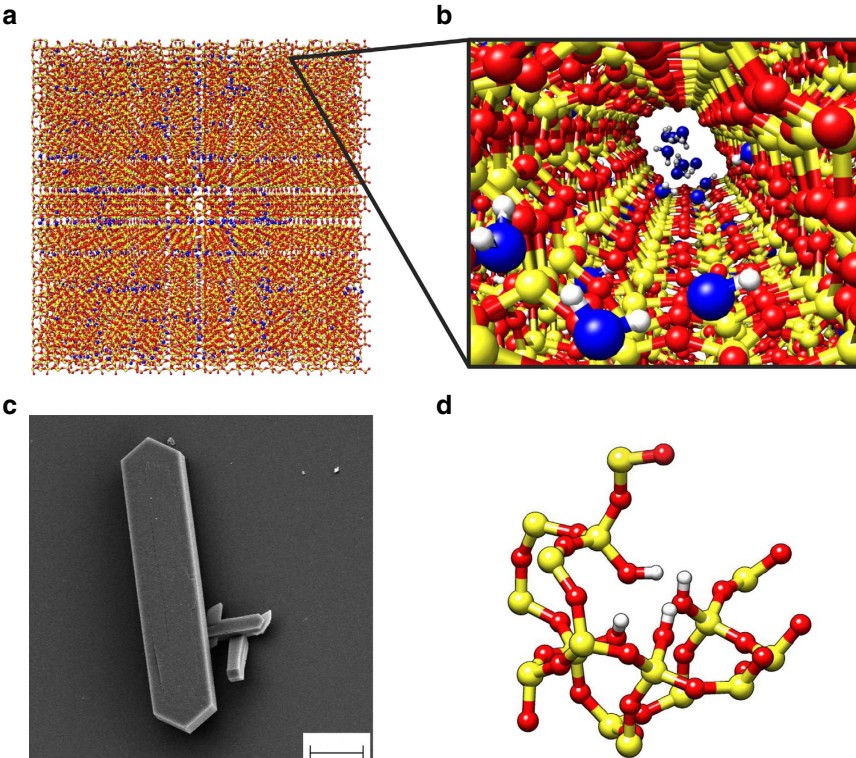

**Figure 1 | MD and experimental specimens of zeolite.** (**a**) Periodic silicalite-1 crystal (red/yellow) infiltrated by water molecules (blue) for a studied MD setup. (**b**) Detail of the water intrusion in a silicalite-1 pore. (**c**) Scanning electron microscopy analysis of a silicalite-1 crystal (scale bar, 10 μm). (**d**) 'Silanol nests model' of the hydrophilic defects induced by Al insertion in the silicalite-1 framework (silicon atoms are yellow; oxygen red; hydrogen white)[30]. Rendering MD pictures are made with UCSF Chimera[68].

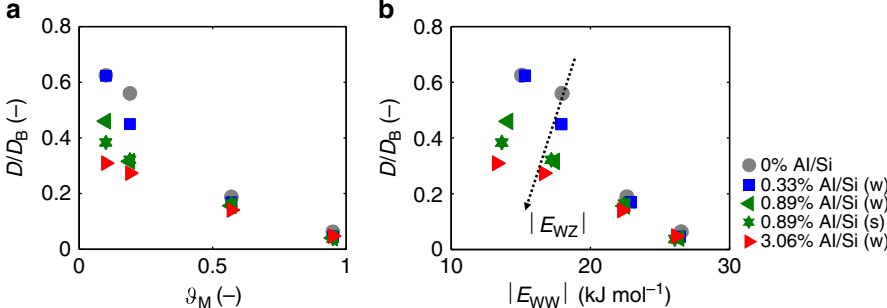

**Figure 2 | Self-diffusivity of water in defected MFI membranes.** (**a**) Self-diffusion coefficient of water ($D$) nanoconfined in defected zeolite crystals. $D$ is normalized by $D_B$ (self-diffusion coefficient of bulk water) and evaluated at different $\vartheta_M$ (pore hydration). (**b**) Self-diffusion coefficient of water nanoconfined in MFI zeolites with various defect concentrations versus water–water-specific interaction energies ($E_{ww}$, see Supplementary Equation (1)). The effect of the water–zeolite-specific interaction energy ($E_{wz}$, see Supplementary Equation (2)) on $D$ is also shown. In the legend, (w) refers to 'weak' silanol defects ($q_H = 0.45e$; $q_O = -0.9e$), whereas (s) to 'strong' ones ($q_H = 0.65e$; $q_O = -1.1e$). According to the specific adsorption-infiltration isotherm (see Supplementary Fig. 2), the considered pore hydration may involve either adsorption or infiltration regimes.

measures are performed[19]. When the experimental measurements probe diffusivities over distances on the order of nanometres (that is, neutron scattering), the results show a reasonable (that is, within an order of magnitude) agreement with those measurements calculated from MD simulations[48,49,53]. However, other measurement techniques (for example, gravimetric uptake quantify or pulsed field gradient nuclear magnetic resonance) evaluate water transport over much larger distances ($\sim 1$–$100\,\mu m$) and these measured diffusivities are generally three to six orders of magnitude lower than those found by molecular simulations or neutron scattering[27,54,55]. This significant decrease in the rate of mass transport over larger distances is also observed with zeolite-based membranes, as the water flux across such

membranes is typically four to eight orders of magnitude lower than what is predicted with MD simulations[42,56]. Therefore, additional transport resistances must exist to provide a physical reasoning for these large discrepancies[19].

Recently, it has been reported that additional transport barriers (arising from contamination, pore blockages or pore collapse) occurring at the surface of zeolite crystals could potentially explain the substantial difference between effective (apparent) and intracrystalline diffusivity of hydrocarbons in zeolites[21,22,57]. When the fluid diffuses through surfaces ('barriers') of dramatically decreased permeability, the overall measured mass transport properties in the porous material may significantly change. Hence, it is possible to define the surface permeability ($\alpha$)

of such barriers as the factor of proportionality between fluid flux and difference in concentrations DONs through the barrier, namely $j_x = -\alpha(c_l - c_r)$, where $c_l$ is the concentration on the left side of the barrier and $c_r$ the concentration on the right one[19].

Some authors have interpreted the surface permeability as a homogeneous layer of zeolite with dramatically reduced diffusivity near particle surface; however, nowadays the most accepted interpretation considers surface barriers as complete blockage (and/or partial narrowing) of the majority ($>99.99\%$) of pore entrances, with the exception of a few accessible pore mouth openings[22,58,59]. In other words, the surface of nanoporous particles can be seen as a generally impermeable layer showing rare and homogeneously dispersed pore openings[21,60]. In case of zeolites, the blockage or the narrowing of pores could arise from a variety of surface defects, mainly either the presence of large amorphous silica surface patches or local surface terminations that block pores' entrance, although no direct structural evidence has been presented so far[57]. This crust is considered as responsible of the mass transport rate limitations experimentally noticed in nanoporous materials. In this sense, the effect of surface permeability can be interpreted as a 'detour' that guest molecules have to take before diffusing into the nanoporous framework, which eventually leads to longer diffusion paths and thus reduced apparent diffusivities[22]. On the one hand, the magnitude of these detours is determined by surface characteristics rather than by the peculiar mechanisms of molecular transport; on the other hand, the effect of surface detours on the effective (apparent) diffusion is more prominent in smaller nanoporous particles, where the length of such additional surface diffusion paths becomes comparable with intracrystalline ones[3,57].

To analyse the effect of the surface barriers on the overall transport behaviour, we first developed a one-dimensional diffusion model with an additional 'surface resistance' term and then experimentally quantified this term through supplementary uptake experiments.

**Diffusion resistance model**. As schematically shown in Fig. 3c, water molecules are subject to a series of diffusion resistances, while diffusing through the zeolite specimen: (1) starting from bulk conditions, the water molecule (2) 'sticks' to the zeolite surface, then (3) enters in an open pore and (4) diffuses through it. Sticking probability (probability for a fluid molecule to adsorb to a solid surface on colliding with it)[3,17,61] and surface barriers

(planes of dramatically reduced permeability, for example, external pore blockage or narrowing) can be generally considered as complementary phenomena in determining the overall surface diffusion resistance of fluid molecules entering nanopores. Therefore, the series of resistances encountered by water molecules diffusing through the zeolite can be reduced to: (i) a pore entrance resistance $1/\alpha$; (ii) an intracrystalline diffusion resistance $L/D$, where $L = V/S_p$ is the thickness of the specimen (characteristic intracrystalline diffusion length), $S_p$ is the surface where water molecules can intrude nanopores and $V$ is the overall volume of the specimen (see Fig. 3a,b)[19,39,40]. By considering low hydration regimes, equation (2) reduces to $D \cong D_0$ and thus equation (1) can be written as

$$D_T \cong \Gamma D. \qquad (3)$$

Equation (3) only takes into account intracrystalline diffusion resistances, therefore considering a negligible contribution of surface resistances to the overall mass transport of water throughout the nanoporous material. Such assumption is generally valid in case of large crystals (that is, high $V/S_p$ values). However, surface diffusion resistances may have a fundamental influence in the case of submicrometre crystals, which can strongly reduce the effective diffusivity ($D_{eff}$) experimentally measured by sorption rate measurements. Hence, surface effects on water diffusivity can be introduced in equation (3) as

$$D_{eff} \cong \Gamma \left( \frac{\Gamma}{\alpha V / S_p} + \frac{1}{D} \right)^{-1}, \qquad (4)$$

where all surface resistances have been incorporated in the $1/\alpha$ term, as a first approximation (see Supplementary Note 3 for further details)[62]. Consequently, in small zeolite crystals ($V/S_p \to 0$), the diffusion resistance due to surface barriers tends to large values and $D_{eff}$, correspondingly, approaches zero. At the other extreme, that is, in systems where the volume-to-surface ratio of the membrane is large, surface diffusion resistance is negligibly small compared with intracrystalline one; therefore, $D_{eff} \cong \Gamma D_0$.

Although $D$ is a property of nanoconfined water, $\alpha$ is related to the real characteristics of the experimental crystal (for example, surface defects and pore blockage). These surface barriers can depend strongly on the conditions under which the zeolite has been synthesized, stored and prepared for measurement[63].

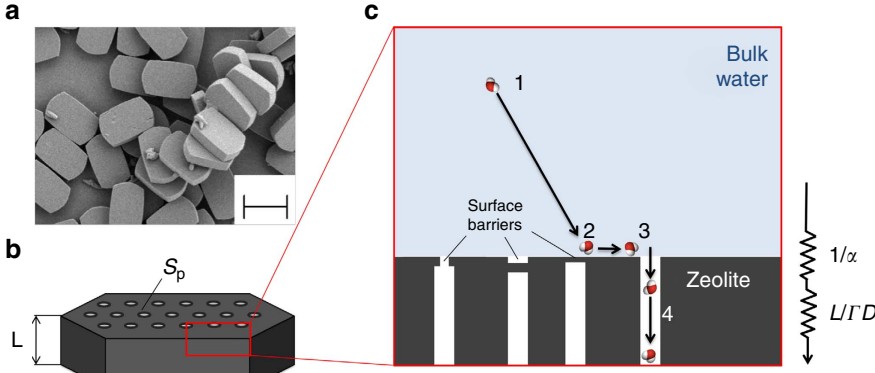

**Figure 3 | Diffusion resistances for water at the nanoscale. (a)** Scanning electron microscopy of the considered MFI zeolites (scale bar, 3 μm). **(b)** Representation of a zeolite specimen: water molecules can intrude the nanopores through the surface $S_p$, whereas $V = S_p L$ is the volume of the specimen. **(c)** Water transport through microporous materials is influenced by a series of nanoscale mass transport phenomena, namely: (1) bulk diffusion; (2) sticking probability to the solid surface; (3) resistance to micropore intrusion due to surface barriers ($1/\alpha$); (4) diffusion resistance in the micropore network ($L/(\Gamma D)$).

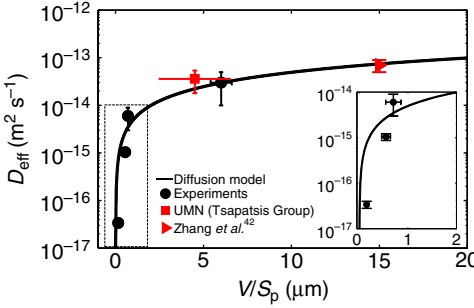

**Figure 4 | Experimental diffusion coefficient.** Effective diffusion coefficient ($D_{eff}$) of water in silicalite-1 specimens with different volume/accessible surface ratio ($V/S_p$). Experimental results (symbols) are fitted by the diffusion resistance model (black line) in equation (4) ($\alpha = 5.0 \times 10^{-9}$ m s$^{-1}$; coefficient of determination $R^2 = 0.94$). Error bars are given by uncertainties in the determination of $V/S_p$ by imaging and $D_{eff}$ by water uptake experiments (s.d., 98% confidence interval). It is noteworthy that $D \cong 0.6 D_B = 1.37 \times 10^{-9}$ m$^2$ s$^{-1}$ (from MD simulations of water within silicalite-1 in the adsorption regime, with $D_B = 2.30 \times 10^{-9}$ m$^2$ s$^{-1}$ experimentally measured for water at 300 K (ref. 69)) and $\Gamma = 1.11$ (from experimental adsorption isotherms of water in silicalite-1 (ref. 27)) were considered for the model fitting. In the inset, the significant effect of surface barriers at low $V/S_p$ is highlighted. Red symbols represent experimental water diffusion in silicalite-1 samples reported in the literature (triangle[42]) or provided by Michael Tsapatsis, University of Minnesota (square).

**Experimental observations**. The influence of surface barriers on the effective diffusivity of water was then investigated by experimentally measuring the diffusivity in purely siliceous MFI crystals with varying size (that is, as a function of the characteristic length $L = V/S_p$, see also Supplementary Fig. 8). The results of these experiments are shown in Fig. 4.

Figure 4 demonstrates that the effective diffusivity of water measured in zeolite crystals decreases substantially as the crystal size decreases (our previous work showed that the effective defect density of these crystals is approximately equal[27]). Using equation (4) and MD results for self-diffusivity, we obtained $\alpha = 5.0 \times 10^{-9}$ m s$^{-1}$, which is similar to values reported for other molecule/zeolites systems in the literature (see Supplementary Note 4 for quantitative comparisons with other works). These results indicate that, in particular for thin zeolite samples the surface resistance becomes the limiting resistance and substantially decreases the mass transfer through these materials. Consequently, together with intracrystalline diffusivity and membrane solubility, $\alpha$ is the third fundamental parameter to be considered in the design of nanoporous membranes for RO applications with enhanced permeability.

To show the impact of surface barriers on membrane permeability, we consider the transport diffusivities of water in MFI zeolites with increasing Al substitutions and thus hydrophilicity, as experimentally found in our previous work[27]. In Fig. 5, both water self-diffusivities from MD simulations (red triangles, see also Supplementary Fig. 9) and experimental corrected diffusivities (black dots) show a decreasing trend with framework hydrophilicity; however, a six orders of magnitude difference is noticeable for the absolute values, even though $D_{eff} \cong D_T$ should approximate $\Gamma D$ at low pore hydrations (equation (3)). On the contrary, if the self-diffusivities by MD are introduced in equation (4) and thus both intracrystalline and surface effects are taken into account, the resulting corrected diffusivities $\left(\frac{\Gamma}{\alpha \frac{V}{S_p}} + \frac{1}{D}\right)^{-1}$ (blue triangles, where $\alpha = 5.0 \times 10^{-9}$ m s$^{-1}$ is assumed as constant and equal to the experimental value found

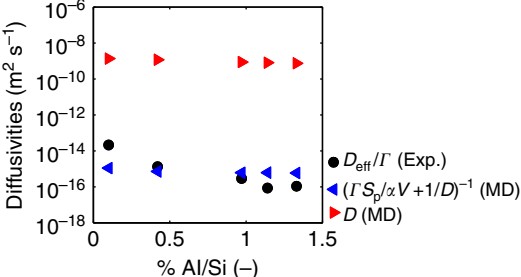

**Figure 5 | Surface effects on water diffusion.** Diffusion coefficients of water in MFI zeolites from experiments (dots) and model (triangles) with increasing concentration of hydrophilic defects. Self-diffusivity ($D$, red triangles) is obtained from MD simulations, averaged in the adsorption range and rescaled by the experimental self-diffusivity of bulk water at 300 K (see Supplementary Fig. 9). Corrected diffusivities are obtained by either considering the effect of surface barriers on MD results by equation (4) ($\left(\frac{\Gamma}{\alpha \frac{V}{S_p}} + \frac{1}{D}\right)^{-1}$, blue triangles, where $\alpha = 5.0 \times 10^{-9}$ m s$^{-1}$ is assumed as constant and equal to the experimental value found for silicalite-1) or by experiments ($D_{eff}/\Gamma$, black dots; see Fig. 4 and Supplementary Fig. 2 in ref. 27).

for silicalite-1 as a first approximation) are of the same orders of magnitude of the experimental ones. Moreover, it can be argued that the remaining differences between experimental diffusivities and self-diffusivities by MD corrected by equation (4) are due to the first approximation value of $\alpha$. In general, $\alpha$ depends on the synthesis, storage and pretreatment of the considered sample, as well as on the adopted experimental protocol[16,17], which were different from the experiments considered here. However, by fitting the experimental $D_{eff}/\Gamma$ values reported in Fig. 5 with equation (4), where $D$ come from MD and $V/S_p$ are experimentally measured[27], the fitted $\alpha$-values show an exponentially decreasing trend with the defect concentration (see Supplementary Fig. 10). This result suggests that Al defects are not only limited to inner crystals but also surface defects are introduced, which cause an exponential decrease in surface permeability and thus effective diffusivity (see Supplementary Note 5).

In conclusion, within the class of nanoporous materials, zeolites have been trending towards active layer thicknesses approaching a few nanometres to reduce intracrystalline diffusion resistance; however, these nanometric length scales did not lead to faster transport time as expected[57]. Even though other phenomena may be also involved (for example, concentration polarization), surface barriers may have a paramount importance in limiting the diffusivity of guest molecules and thus the permeability of membranes with nanometric thickness: equation (4) predicts a linear increase of diffusivity, and thus membrane permeability, with $\alpha$. For example, the silicalite-1 membranes experimentally tested in our previous work[27] may show a tenfold permeability enhancement by reducing ten times the amount of surface imperfections, namely from 40 to 400 Barrer (see Supplementary Fig. 11 and Supplementary Note 6).

## Discussion

The mass transport of guest molecules is essential to determining the performance of nanoporous materials in several technological sectors. However, a complete theoretical understanding of such transport phenomena is still incomplete and orders of magnitude discrepancies between simulations and experiments are often reported in the literature. The novel experimental evidences, diffusion resistance model and hybrid experimental-simulation approach reported in this work may represent further puzzle pieces in understanding and predicting diffusion resistances

through nanoporous materials, with particular focus on water sorption into MFI zeolites for desalination.

In particular, the evidence that surface barriers have a fundamental role in limiting the guest molecules transport in zeolites is found to be true not only for hydrocarbons (as already demonstrated in the literature) but also for water (a much smaller molecule); thus, both the intracrystalline and surface diffusion resistances dictate the effective diffusivity of water molecules in MFI zeolites. First, MD simulations were performed to understand the decrease in intracrystalline water diffusion in MFI zeolites with increasing hydrophilicity. Second, water uptake experiments were performed on pristine MFI zeolite samples with decreasing volume to surface ratio, to evaluate the effect of surface barriers (for example, surface pore blockage or narrowing) on the effective diffusivity. A simple diffusion resistance model was then proposed and validated by experiments. The results demonstrated that the effect of the surface barriers significantly decreased the rate of mass transport through the zeolite crystals. Such an outcome should not be underestimated, because zeolites have long been proposed as an ideal candidate to replace the active layer in RO membranes used for water desalination. However, thus far, the experimentally measured permeability of water through zeolite-based membranes is typically one to two orders of magnitude lower than state-of-the-art polymeric membranes, whereas previous molecular simulations predicted at least an order of magnitude increase in water permeability. However, all previous molecular simulations have assumed that the zeolite is a perfectly crystalline material and have neglected any mass transport resistances at the interface. Hence, we expect that this article may drive the attention of designers of novel membranes—especially for desalination and molecular sieving or sensing applications[64]—to the reduction of surface barriers (for example, by means of recent microimaging-controlled removal techniques), which has the potential to enhance their permeability by orders of magnitude.

## Methods

**MD simulations.** To ensure good statistics at low water fillings, the MFI structures consist of $5 \times 5 \times 4$ crystal cells ($10.0 \times 9.9 \times 5.4$ nm$^3$; Fig. 1a). Simulations were carried out considering periodic conditions at the $x, y, z$ boundaries. The molecular interactions included in the MD force field were (i) bonded (intramolecular) interactions, described by harmonic potentials; and (ii) non-bonded (intermolecular) interactions, where 12-6 Lennard–Jones and Coulomb potentials were adopted to model van der Waals and electrostatic interactions, respectively. A detailed discussion on the MD parameters and force field is reported in the Supplementary Methods.

The initial configurations were relaxed by energy minimization and the resulting equilibrium coordinates were restrained by harmonic potentials. After the initialization of velocities according to Maxwell distribution, the configuration was then simulated in canonical ensemble with constant number of particles (N), system's volume (V) and absolute temperature (T) also known as NVT ensemble ($T = 300$ K, $t = 100$ ps)[65]. The equilibrated structure was successively hydrated by randomly inserting water molecules in the zeolite pore volumes, by means of a Monte Carlo-like algorithm[66]. The amount of introduced solvent molecules per unit cell was designed to exploring the whole range of pore filling ($\omega = 5,10,30,50$ N/UC) for pressures at both adsorption and infiltration regimes (see Supplementary Figs 2 and 3). The hydrated structure was again equilibrated by energy minimization and NVT ensemble ($T = 300$ K, $t = 300$ ps) simulations[65]. Finally, production runs were carried out in the canonical ensemble ($T = 300$ K and thermostat time constant $\tau = 0.1$ ps (ref. 67)) for up to 2 ns to reach the equilibrium state. The self-diffusivity ($D$) of inner water was finally determined from the Einstein relation at steady-state conditions and the convergence of the obtained values during the simulated trajectory was checked.

**Experimental materials and methods.** The synthesis procedure and characteristics of the MFI zeolites experimentally tested are reported in our previous works[27,44]. The zeolites are named according to their respective $V/S_p$, which was obtained from imaging (see Fig. 1 in ref. 44). Physisorption of water vapour was carried out according to the procedure reported in ref. 27 (see Supplementary Fig. 8). The diffusivity was then estimated fitting results to Fick's law (see equation (1) in ref. 27) and thermodynamically corrected by equation (1).

**Data availability.** The data that support the findings of this study are available from the corresponding authors upon request.

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

### Acknowledgements

We are grateful to MITOR project (Compagnia di Sanpaolo) for travel support. P.A., E.C. and M.F. acknowledge the NANO-BRIDGE (PRIN 2012, grant number 2012LHPSJC) and NANOSTEP (Fondazione CRT, Torino) projects. M.F. acknowledges travel support from the Scuola Interpolitecnica di Dottorato—SCUDO. M.F., P.A. and E.C. acknowledge the CINECA award under the ISCRA initiative, for the availability of high-performance computing resources and support (Iscra C projects DISCALIN and COGRAINS, PRACE project MULTINANO). M.F., P.A. and E.C. acknowledge also the computational resources provided by HPC@POLITO (http://www.hpc.polito.it). T.H. and E.N.W. acknowledge MIT-KFUPM Center for Clean Energy and Water that supported the experimental work.

### Author contributions

T.H., E.N.W. and M.T. carried out the experimental part of this work. M.F., A.B., E.C. and P.A. carried out the modelling part of this work. All authors helped in writing the paper and commented on it.

### Additional information

**Competing financial interests:** The authors declare no competing financial interests.

