## [Peer Review File · Nature Communications]

Reviewers' Comments:

Reviewer #1 (Remarks to the Author)

Referee Report NCOMMS-16-03233 on "Interplay between hydrophilicity and surface barriers ..." by M. Fasano et al. in Nature Communications

The paper deals with a hot topic of both fundamental and applied research and is, owing to its rather broad relevance, doubtlessly of particular interest for the readership of Nature Communications. Though over already decades diffusion is known to be key to many phenomena occurring in nature and technology, it is only the most recent development of sophisticated measuring techniques which has, eventually, paved the way towards exploring the relevant, process-governing steps in mass transfer. This is in particular true with nanoporous materials. In addition to their relevance in fundamental research as host-guest systems with essentially universally variable properties, manifold technological applications have also made them high-tech materials. Most unexpectedly, in such systems in numerous cases the permeation through the particle surface has been found to dominate over the influence of the diffusional resistance of the guest molecules in the intracrystalline space. The very mechanism giving rise to these resistances are, however, still far from being understood and there is urgent need for a thorough investigation of mass transfer phenomena at the interface between nanoporous materials and the surroundings. The present paper is devoted to exactly this goal.

With zeolites of type MFI, the authors focus their studies on a host system which, owing to its relevance in heterogeneous catalysis, is among the most often investigated nanoporous materials. With water they do, moreover, consider a host molecule which, given its omnipresence in nature and technology, is doubtlessly a good choice for ensuring a widespread benefit of the reported studies and findings in other fields. The paper is centered around the performance of MD simulations of water diffusivities in MFI as a function of the defect concentration, with the distinction between "weak" and "strong" ones. For correlating these data with the results of transient uptake experiments, the authors considered superimposed effects of diffusion resistances in the bulk (reciprocal value of the intracrystalline diffusivity times crystal height) and on the surface (reciprocal surface permeability) where, most astonishingly, good agreement was already obtained by considering a uniform surface permeability. Finally, as a particular highlight, it is even possible to demonstrate satisfactory agreement between the (effective) diffusivities as resulting in uptake experiments and their equivalent resulting by appropriately (namely, by their eq. (4)) combining the simulated diffusivity data with the postulated surface resistance, on considering a large variety of samples with different defect concentrations.

From my perspective, the paper might, essentially, be presented as it stands. I would, however, suggest that the authors consider a reformulation of the legend to Fig. 3. The sticking probability (p_s) does, obviously, not appear in their eq. (4) and this is completely correct. In this context it is, possibly, useful to refer to the two possible meanings associated with the term "sticking probability". The first, conventional one refers to mere surfaces and indicates the probability that, upon colliding with the surface, a molecule remains stuck to the surface. The similar question may clearly also be asked with respect to the external surface of a nanoporous particle. However, here one has to put the question in a different way, namely whether, upon colliding with the surface of a nanoporous particle, a molecule is able to overcome the barrier and to get into the genuine pore space. This probability (also referred to as the "sticking coefficient" of nanoporous materials) is - as evidenced, e.g., in their reference [54] - proportional to the surface permeability. Or saying it in other words: it is nothing else than the surface barrier (as appearing in terms of the reciprocal surface permeability) which prevents, upon colliding with the external surface, the molecules from propagating into the genuine pore space. If there is no space for referring to this quite complex Situation extensively, it were completely sufficient to refer in parentheses to the fact that the effect of step 2 is incorporated into the magnitude of the surface permeability α (or to simply omit this step 2 in the sequence, considering it as being incorporated in (the present) step 3)

Reviewer #2 (Remarks to the Author)

This manuscript is a combination of experiments and molecular simulations designed to understand transport phenomena in zeolites. It deals with a long standing problem that diffusivities observed from molecular simulations in a model perfect crystals are usually several orders of magnitude larger than the effective diffusivity observed in experiments. One of the working hypotheses for slow diffusion of water in zeolites is the existence of hydrophilic defects within the zeolite structure.

The authors address these issues in a two-prong attack. Using molecular simulations they investigate how self-diffusion coefficient inside zeolites depends on the number of hydrophilic defects (represented by Al atoms). They show that presence of internal defects alone cannot explain the differences in diffusivity. Now, building on the alternative hypothesis that the resistance to the mass transfer is concentrated on the surface of the zeolite crystal, they carried out a set of experiments measuring diffusion in zeolites of different size and V/S ratio. This allowed them to extract value of the term related to surface permeability (α). A model that combines surface resistances and intracrystalline resistances (as measured from molecular simulations) matches experimental results very well.

The molecular simulation aspect of the work explained clearly. The article is sound in terms of methodology, clarity and robustness of the conclusions. Few occasional typos can be weeded out during the review process.

Where I believe article stops short from providing an important insight is the nature of the surface resistances on molecular level. Is it something that can also be modelled/predicted from appropriate models to make the computational approach fully predictive? Is it an activated process, so some additional information on the magnitude of the barriers can be extracted from the dependency of transport on temperature?

So my recommendation is to publish the article with an additional comment from the authors on the nature of α

Reviewer #3 (Remarks to the Author)

This paper highlights the importance of surface barriers in the diffusion of water in hydrophilic membranes. The topic is obviously very important as outlined in the introduction, experimental diffusivities, especially those measured in transport experiments, are generally much lower than theoretical simulations predict. The quality of the paper is high, it is written in a scholarly manner, accessible and complete.

The work shows that by decreasing the crystal size of the zeolite the diffusion decreases hinting at the importance of barriers. I am not an expert in this field but I do not think this supposition in itself is completely novel and indeed the paper cites several works referring to this resistance at the surface.

With this in mind, the main problem of this paper is that it does not provide novel insight into the nature of these surface barriers. The MD simulations pertain to internal diffusion in micropores. Although the results of the theoretical study in this work are good (a critical note on the model is the choice to represent acidic bridging hydroxyl groups as silanol defects; this is in my view not necessary [Al can be incorporated easily] and also incorrect as it overestimates the hydrophilicity!), they turn out to be not very relevant for the point to be made.

Experimentally, it is shown that some effect of the crystal surface where the molecules absorb to enter the micropores cause a high resistance. As said, this is not new and the paper does not provide clear insight. Evenmore, the conclusions seem to be that more work is needed to

understand this effect.

So the authors should make better clear where the novelty lies, otherwise I feel this work is not suitable for Nature Communications.

REVIEWER #1

Q1.1 The paper deals with a hot topic of both fundamental and applied research and is, owing to its rather broad relevance, doubtlessly of particular interest for the readership of Nature Communications. Though over already decades diffusion is known to be key to many phenomena occurring in nature and technology, it is only the most recent development of sophisticated measuring techniques which has, eventually, paved the way towards exploring the relevant, process-governing steps in mass transfer. This is in particular true with nanoporous materials. In addition to their relevance in fundamental research as host-guest systems with essentially universally variable properties, manifold technological applications have also made them high-tech materials. Most unexpectedly, in such systems in numerous cases the permeation through the particle surface has been found to dominate over the influence of the diffusional resistance of the guest molecules in the intracrystalline space. The very mechanism giving rise to these resistances are, however, still far from being understood and there is urgent need for a thorough investigation of mass transfer phenomena at the interface between nanoporous materials and the surroundings. The present paper is devoted to exactly this goal.

R1.1 We thank the reviewer for these comments. We have added some further references and discussions in the text to highlight the broad variety of fields where the new perspectives presented by this work may find application, for instance the exponentially growing desalination industry. ¹

Q1.2 With zeolites of type MFI, the authors focus their studies on a host system which, owing to its relevance in heterogeneous catalysis, is among the most often investigated nanoporous materials. With water they do, moreover, consider a host molecule which, given its omnipresence in nature and technology, is doubtlessly a good choice for ensuring a widespread benefit of the reported studies and findings in other fields. The paper is centered around the performance of MD simulations of water diffusivities in MFI as a function of the defect concentration, with the distinction between "weak" and "strong" ones. For correlating these data with the results of transient uptake experiments, the authors considered superimposed effects of diffusion resistances in the bulk (reciprocal value of the intracrystalline diffusivity times crystal height) and on the surface (reciprocal surface permeability) where, most astonishingly, good agreement was already obtained by considering a uniform surface permeability. Finally, as a particular highlight, it is even possible to demonstrate satisfactory agreement between the (effective) diffusivities as resulting in uptake experiments and their equivalent resulting by appropriately (namely, by their eq. (4)) combining the simulated diffusivity data with the postulated surface resistance, on considering a large variety of samples with different defect concentrations.

R1.2 We are glad for the positive feedback from the reviewer. In the revised manuscript, we provided a better argumentation of both Eq. (4) and the physical meaning of surface permeability (see Supplementary Notes 3-5), in order to improve the mechanistic understanding of surface diffusion resistance observed in our experiments.

Q1.3 From my perspective, the paper might, essentially, be presented as it stands. I would, however, suggest that the authors consider a reformulation of the legend to Fig. 3. The sticking probability (p_s) does, obviously, not appear in their eq. (4) and this is completely correct. In this context it is, possibly, useful to refer to the two possible meanings associated with the term "sticking probability". The first, conventional one refers to mere surfaces and indicates the probability that, upon colliding with the surface, a molecule remains stuck to the surface. The similar question may clearly also be asked with respect to the external surface of a nanoporous particle. However, here one has to put the question in a different way, namely whether, upon colliding with the surface of a nanoporous particle, a molecule is

able to overcome the barrier and to get into the genuine pore space. This probability (also referred to as the "sticking coefficient" of nanoporous materials) is – as evidenced, e.g., in their reference [54] - proportional to the surface permeability. Or saying it in other words: it is nothing else than the surface barrier (as appearing in terms of the reciprocal surface permeability) which prevents, upon colliding with the external surface, the molecules from propagating into the genuine pore space. If there is no space for referring to this quite complex Situation extensively, it were completely sufficient to refer in parentheses to the fact that the effect of step 2 is incorporated into the magnitude of the surface permeability Alpha (or to simply omit this step 2 in the sequence, considering it as being incorporated in (the present) step 3) .

R1.3 As correctly highlighted by the reviewer, sticking probability (probability for a fluid molecule to adsorb to a solid surface upon colliding with it) and surface barriers (planes of dramatically reduced permeability, e.g. external pore blockage) can be generally considered as complementary phenomena in determining the overall surface diffusion resistance of fluid molecules entering nanopores.

Hence, in the qualitative schematics reported in Fig. 3, we considered them as distinct phenomena in a series of diffusion resistances: (1a) the water molecule “sticks” to the zeolite surface, then (1b) enters in an open pore and (2) diffuses through it. However, current experiments and simulations do not allow clearly distinguishing between sticking and surface barriers diffusion resistances; therefore, we chose to incorporate both effect in the surface permeability (α), as detailed in Supplementary Note 3.

To make this concept clearer to the reader and in accordance with the reviewer’s suggestion, in the revised version of the manuscript we improved Fig. 3 as well as some parts of the main and supplementary text.

REVIEWER #2

Q2.1 This manuscript is a combination of experiments and molecular simulations designed to understand transport phenomena in zeolites. It deals with a long standing problem that diffusivities observed from molecular simulations in a model perfect crystals are usually several orders of magnitude larger than the effective diffusivity observed in experiments. One of the working hypotheses for slow diffusion of water in zeolites is the existence of hydrophilic defects within the zeolite structure.

The authors address these issues in a two-prong attack. Using molecular simulations they investigate how self-diffusion coefficient inside zeolites depends on the number of hydrophilic defects (represented by Al atoms). They show that presence of internal defects alone cannot explain the differences in diffusivity. Now, building on the alternative hypothesis that the resistance to the mass transfer is concentrated on the surface of the zeolite crystal, they carried out a set of experiments measuring diffusion in zeolites of different size and V/S ratio. This allowed them to extract value of the term related to surface permeability (α). A model that combines surface resistances and intracrystalline resistances (as measured from molecular simulations) matches experimental results very well.

R2.1 We are glad that the reviewer remarks both the relevance of the problem and the original approach adopted in this work. Further considerations on the original contribution and perspectives outlined by this article have been introduced in the main text.

Q2.2 The molecular simulation aspect of the work explained clearly. The article is sound in terms of methodology, clarity and robustness of the conclusions. Few occasional typos can be weeded out during the review process.

R2.2 We thank the reviewer for these comments. We have extensively reread the manuscript and rephrased some paragraphs to improve the clarity of the text.

Q2.3 Where I believe article stops short from providing an important insight is the nature of the surface resistances on molecular level. Is it something that can also be modelled/predicted from appropriate models to make the computational approach fully predictive? Is it an activated process, so some additional information on the magnitude of the barriers can be extracted from the dependency of transport on temperature? So my recommendation is to publish the article with an additional comment from the authors on the nature of α .

R2.3 We thank the reviewer for having stimulated this fruitful discussion. The following comments have been integrated in the main text and Supplementary Notes 3-5 to further elaborate on the nature of surface permeability.

Effective diffusion model. The basis of equation (4) in the main text is depicted by the schematic in **Fig. 3c**, where water molecules are subject to a series of diffusion resistances while diffusing through the zeolite specimen: (1) starting from bulk conditions, the water molecule (2) “sticks” to the zeolite surface, then (3) enters in an open pore and (4) diffuses through it. Sticking probability (probability for a fluid molecule to adsorb to a solid surface upon colliding with it) and surface barriers (planes of dramatically reduced permeability, e.g. external pore blockage or narrowing) can be generally considered as complementary phenomena in determining the overall surface diffusion resistance of fluid molecules entering nanopores (diffusion steps 2 and 3 in **Fig. 3c**). However, current experiments

and simulations do not allow to clearly distinguish between sticking resistance (which has not a general, analytical formulation) and surface barriers diffusion resistances; therefore, in equation (4) in the main text we chose to approximate the series of diffusion resistances as (i) surface barrier resistance and (ii) intracrystalline (volumetric) resistance.

First, let us recall the definition of surface permeability (α), namely the proportionality factor between fluid flux and difference in concentrations on either sides of a surface barrier:^{2,3}

$$j_x = -\alpha(c_l - c_r), \quad (\text{R1})$$

being j_x the fluid flux through the barrier, c_l and c_r concentrations on the left and right sides of the barrier, respectively. Second, Fick's law relates the diffusive flux of fluid to a concentration gradient at steady state:

$$J = -D_T \nabla c, \quad (\text{R2})$$

being D_T the Fick's diffusion coefficient and c the fluid concentration. In zeolite pores, fluid transport can be approximated as a one-dimensional diffusion (*e.g.*, along x axis); therefore, equation (R2) can be reduced to

$$j_x = -D_T \frac{\partial c}{\partial x}, \quad (\text{R3})$$

where $\frac{\partial c}{\partial x}$ refers to the concentration gradient along the pore length.

Let us now consider the transport of water molecules from bulk conditions (step 1 in **Fig. 3c**, fluid concentration c_1) into zeolite framework (step 4 in **Fig. 3c**, fluid concentration c_3). Hence, the water flux through the zeolite sample (j_x) during the uptake process can be expressed by equations (R1) and (R3) as

$$j_x = -\alpha(c_1 - c_2) \quad (\text{R4})$$

and

$$j_x = -D_T \frac{(c_2 - c_3)}{L}, \quad (\text{R5})$$

respectively, where $c_1 = c_l$, $c_2 = c_r$ and L is the pore length. By imposing the continuity of j_x through the zeolite pores, Equations (R4) and (R5) can be then combined as:

$$j_x = -\left(\frac{1}{\alpha} + \frac{L}{D_T}\right)^{-1} (c_1 - c_3). \quad (\text{R6})$$

Hence, the water transport into zeolite pores is determined by an effective diffusion resistance ($R_{eff} = L/D_{eff}$), which – analogously to the lumped element models adopted for electric or thermal applications – arises from a series of surface and volume resistances to diffusion, namely

$$R_{eff} = \frac{1}{\alpha} + \frac{L}{D_T}. \quad (\text{R7})$$

At low hydration regimes, Barrer approximation holds and thus $D_T \cong \Gamma D$ (*i.e.*, Darken's equation ²), where Γ is the thermodynamic factor and D the self-diffusion coefficient. The water transport in uptake experiments can be finally expressed in terms of the effective diffusivity (D_{eff}), that is

$$D_{eff} \cong \left(\frac{1}{\alpha L} + \frac{1}{\Gamma D} \right)^{-1} \quad (\text{R8})$$

Multiplying by Γ both numerator and denominator and recalling that $L = V/S_p$ in the considered zeolite samples yield finally to the equation (4) reported in main text, namely

$$D_{eff} \cong \Gamma \left(\frac{\Gamma}{\alpha V/S_p} + \frac{1}{D} \right)^{-1} \quad (\text{R9})$$

Mechanistic interpretation. Some authors have interpreted the surface permeability as a homogeneous layer of zeolite with dramatically reduced diffusivity in the proximity of particle surface; however, nowadays the most accepted interpretation considers surface barriers as complete blockage (and/or partial narrowing) of the majority (>99.99%) of pore entrances, with the exception of a few accessible pore mouth openings. ⁴⁻⁶ In other words, the surface of nanoporous particles can be seen as a generally impermeable layer showing rare and homogeneously dispersed pore openings. ^{7,8} In case of zeolites, pore blockage or narrowing could arise from a variety of surface defects, mainly either the presence of large amorphous silica surface patches or local surface terminations that block pores' entrance, although no direct structural evidence has been presented so far. ⁹

This crust is considered as responsible of the mass transport rate limitations experimentally noticed in nanoporous materials. In this sense, the effect of surface permeability can be interpreted as a "detour" that guest molecules have to take before diffusing into the nanoporous framework, which eventually leads to longer diffusion paths and thus reduced apparent diffusivities. ⁵ On the one hand, the magnitude of these detours is determined by surface characteristics rather than by the peculiar mechanisms of molecular transport; on the other hand, the effect of surface detours on the effective (apparent) diffusion is more prominent in smaller nanoporous particles, where the length of such additional surface diffusion paths becomes comparable with intracrystalline ones. ^{9,10}

Comparison with literature results. To the best of our knowledge, this is the first study quantifying the magnitude of surface permeability in case of water transport through MFI zeolites. Nevertheless, the resulting α can be compared with previous works in the literature where zeolites (or Metal-Organic Frameworks, MOFs) were typically infiltrated by light hydrocarbons. For example, Chemelik *et al.* reported α in the range 10^{-8} to 10^{-6} m/s for the uptake of methanol in ferrierite zeolite and 10^{-9} to 10^{-7} m/s for propane in MOF, according to different relative concentrations; ¹¹ similar results were also found by Heinke *et al.* in case of methanol uptakes into ferrierite. ¹² Recently, Saint Remi *et al.* experimentally evaluated α and D for commercial zeolites (SAPO-34) intruded by methanol: while α spreads over almost two orders of magnitude (*i.e.*, from 10^{-9} to 10^{-7} m/s), D shows substantially unchanged values. ⁷

The large variety of reported α values can be mainly attributed to the different synthesis, storage and experimental conditions to which are subjected zeolite/MOF samples. In particular, a progressive degradation of surface permeability with storage time was found in several nanoporous samples. For example, Chemelik *et al.* noticed a 5-fold decrease (from 11.2×10^{-9} m/s to 2.3×10^{-9} m/s) in the surface permeability of propane in MOF Zn(tbip) due to an increased storage time of the sample in

ambient atmosphere;¹³ similarly, Guendré *et al.* reported up to one order of magnitude drop in uptake kinetics with storage time (cyclohexane uptake in silicalite-I).¹⁴ This “ageing” effect of zeolite/MOF surfaces is particularly pronounced in case of expositions to water-containing environments, because of the high reactivity of water molecules with the incomplete terminals of zeolite/MOF surfaces.^{13,15,16} These evidences are in line with the relatively low α values reported in our work, where MFI samples were totally immersed in water during uptake experiments.

Despite surface barriers are a peculiar property of each zeolite sample, recent works have demonstrated that surface permeation and intracrystalline diffusion are governed by identical fundamental transport mechanism.⁸ In fact, both surface and volumetric transport phenomena were observed to show the same activation energy, being α/D independent from guest molecules type, loading and equilibrium/non-equilibrium conditions.^{4,5} These observations actually led to the current interpretation of surface barriers as pore blockage phenomena. Furthermore, α has revealed dependence with environment pressure and post-synthesis treatments (e.g., etching), which may both alter the amount of pore mouth openings.^{7,17}

Pore blockage modeling. In contrast with the analysis of water diffusion within the regular network of zeolite’s nanopores, ideal surface structures cannot be defined *a priori* without some arbitrariness. Current experimental techniques are unable to fully characterize the exact nature of surface terminations responsible of surface diffusion resistances, therefore limiting the possibility to compute α by mechanistic considerations. Furthermore, surface barriers can depend strongly on the conditions under which zeolites have been synthesized, stored, prepared for measurement and even on the permeation measurements themselves.^{10,18} To interpret the experimental/numerical values of α , some authors have suggested empirical models based on the probability of both channel mouth opening and intracrystalline channel connection.¹⁰ These models are typically tailored for particular sets of experiments, and require parameters to be empirically fitted^{5,9,19}; hence, a rigorous derivation of predictive models for surface permeability remains an open issue and it is beyond the scope of this work.^{7,10}

However, by considering the water uptake experiments in MFI zeolites by Humplik and colleagues (D_{eff} , Γ and V/S_p ²⁰) and the current molecular dynamics results (D), equation (R9) allows to find a relation between surface permeability and defects concentration (%Al/Si). Results (see **Fig. R1**) show that α undergoes an exponential decrease by increasing the concentration of hydrophilic defects in MFI zeolites, namely

$$\alpha = k_1 \exp(k_2 \times \%Al/Si), \quad (R10)$$

where $k_1 = 1.50 \times 10^{-7}$ m/s and $k_2 = -4.67$ ($R^2 = 0.97$). In fact, the progressive introduction of hydrophilic defects in MFI zeolites may decrease the fraction of surface pore openings, thus reducing α . Therefore, the observed D_{eff} reduction in water uptake experiments with defective MFI zeolites should be mainly attributed to surface effects rather to volumetric ones. In fact, defect concentrations ranging from 0.1 to 1.3 %Al/Si cause a 100-fold α drop (see **Fig. R1**, as from experiments), while only a 2-fold D decrease (see **Fig. R2**, as from molecular dynamics simulations).

Figure R1.

Figure R2.

Finally, better insights on surface permeability could be hardly obtained by the current capabilities of atomistic simulations, because of the large simulation domains needed to analyze statistically relevant pore blockage phenomena. To name an example, simulating realistic 99.99% pore blockages⁸ would require a zeolite-water interface of $\sim 10000 \text{ nm}^2$, which would lead to 50–100 million atoms simulations and microseconds trajectories. Although such atomistic (or eventually coarse-grained) simulations may actually provide more mechanistic details on the nature of α , their computational burden requires top-tier supercomputers and dedicated research activities, which will be certainly tackled in future works but lie beyond the scope of this article.

REVIEWER #3

Q3.1 This paper highlights the importance of surface barriers in the diffusion of water in hydrophilic membranes. The topic is obviously very important as outlined in the introduction, experimental diffusivities, especially those measured in transport experiments, are generally much lower than theoretical simulations predict. The quality of the paper is high, it is written in a scholarly manner, accessible and complete. The work shows that by decreasing the crystal size of the zeolite the diffusion decreases hinting at the importance of barriers.

R3.1 We thank the reviewer for these comments. We have added further references and discussions to highlight the relevance of the work in a broad variety of fields. Moreover, we have rephrased several paragraphs to improve the clarity of the text.

Q3.2 I am not an expert in this field but I do not think this supposition in itself is completely novel and indeed the paper cites several works referring to this resistance at the surface. With this in mind, the main problem of this paper is that it does not provide novel insight into the nature of these surface barriers. The MD simulations pertain to internal diffusion in micropores. [...] Experimentally, it is shown that some effect of the crystal surface where the molecules absorb to enter the micropores cause a high resistance. As said, this is not new and the paper does not provide clear insight. Even more, the conclusions seem to be that more work is needed to understand this effect. So the authors should make better clear where the novelty lies, otherwise I feel this work is not suitable for Nature Communications.

R3.2 We thank the reviewer for stimulating this discussion, which gave us the opportunity to remark the novel contribution and perspectives presented in this article, and make them clearer to the reader. Starting from the comments below, we have added in the main text (introduction and conclusions) and supplementary information (Supp. Note 6) further references and discussions to highlight the novelty of our work in the broad field of guest transport through nanoporous materials, with particular attention to exponentially-growing applications such as desalination and molecular sieving ones.

State of the art. Diffusion resistances due to surface effects (*i.e.*, surface barriers) have taken the scene in the study of transport phenomena through nanoporous materials in the last few years, since a broader adoption of microimaging techniques to monitor transient guest profiles by experiments.² Thanks to these new observations, it has been possible to demonstrate that the transport resistances encountered by light hydrocarbons through zeolites or metal-organic frameworks are mainly due to surface effects rather than to intracrystalline ones, as it has been erroneously believed for decades. This improved understanding has important implications in traditional applications of nanoporous materials, such as selective adsorption and catalysis.

However, while surface barriers are nowadays generally recognized to be due to surface pore blockage or narrowing (see Reply 2.3 to Reviewer #2), predictive models for surface diffusion resistances are still semi-empirical and controversial (see for example the recent discussion in references^{21,22}). Furthermore, a direct computer simulation of the effect of surface barriers on the overall transport through nanopores is not easily accessible by means of current computational facilities (but it could be in the near future), mainly because of the large simulation domains involved (approximately 100 million atoms, see Reply 2.3 to Reviewer #2). To summarize this context, it is helpful to cite the words of Kärger and collaborators about the original contribution of their recent work (*Nature Materials*, April 2016):⁷ “[...] *It is true that our knowledge about the very nature of these surface resistances is*

still rather limited, making it an important subject of current research. However, the detection of a great variability in the strength of these barriers in one and the same batch, as revealed in the present study, may become one of those puzzle pieces which eventually are put together to form a coherent picture. [...]”.

Hence, the new experimental evidences, diffusion resistance model and hybrid experimental-simulation approach reported in this article may represent further puzzle pieces in understanding and predicting diffusion resistances through nanoporous materials, with particular focus on water sorption into MFI zeolites for desalination applications.

Novel contribution. The original contributions of this article are mainly: (i) the evidence of surface barrier phenomena also in case of water uptake into MFI zeolites, which is a fundamental change of perspective for membrane applications; (ii) the quantification of surface diffusion resistances by a diffusion resistance model (Eq. 4 in the main text); (iii) the hybrid experimental-simulation approach to evaluate the effective guest molecules transport through nanoporous membranes.

To the best of our knowledge, this is the first study quantifying the magnitude of surface permeability in case of water transport through MFI zeolites. In fact, while the presence of surface barriers for light hydrocarbons in MFI zeolites was already documented in the literature (see Reply 2.3 to Reviewer #2), this phenomenon had not yet been shown for water molecules guests too. Note that water molecules present a kinetic diameter (≈ 0.26 nm) lower than light hydrocarbons' ones (e.g., ≈ 0.40 nm for methanol).²³ Therefore, the presence of surface barriers on MFI surface (≈ 0.60 nm pore diameter) was not obvious, because their mechanistic nature may involve both total pore blockage (*i.e.*, same effect on guest molecules with different kinetic diameter) and pore narrowing (*i.e.*, different effect on guest molecules with different kinetic diameter). The quantification of surface barriers with water molecules guests has important implications in applications where zeolites have only recently been suggested, such as desalination ones. In fact, desalination industry is expected to show exponential growth in the next decade if innovative membranes can enhance the performances of reverse osmosis plants, thus reducing their operating costs.¹ In this context, zeolite-based membranes have the potential to reject salt ions completely, while permitting water molecules to permeate through with large flow rates.²⁴

However, the experimental permeability of MFI membranes was observed to be orders of magnitude lower respect to theoretical one,²⁰ therefore causing a bottleneck for a possible industrial spreading. While the results of hindered transport are not necessarily new, there is still uncertainty in the literature to explain why this hindered transport occurs (such as internal pore blockages, improper measurement techniques, incorrect parameters for molecular simulations, etc.). However, what we demonstrate in this work is that this hindered transport can be attributed to a surface-based resistance. In particular, the diffusion resistance model reported in Eq. 4, the mechanistic interpretation of α (see Reply 2.3 to Reviewer #2) and the observed relation between α and Si-Al substitutions (see Fig. R1 above and related discussions) are valuable insights for re-focusing the attention of experimentalists on surface barriers reduction in order to enhance the permeability of zeolite-based membranes. In fact, given that membrane permeability is the product of solubility and effective diffusivity of water, solubility is observed to be enhanced by MFI intracrystalline hydrophilicity, whereas effective diffusivity to be decreased.²⁰ However, this work suggests that intracrystalline water diffusivity is relatively little affected by defects introduction (up to 50% reduction respect to pristine crystals, Fig. R2 above), whereas the observed orders of magnitude differences should be ascribed to surface barriers (Fig. R1 above).

These evidences have been achieved by means of an original combination of numerical and experimental analyses, which clearly provide to the reader a comprehensive overview of both volumetric (simulations) and surface (experiments) diffusion mechanisms of water in defective MFI, which are then coupled by the diffusion resistance model (Eq. 4). In fact, molecular dynamics simulations are essential to verify that the experimentally observed reduction in water diffusivity with framework hydrophilicity cannot be explained by intracrystalline diffusion alone, but surface diffusion resistances have a preponderant role. Note that this methodology, which is here adopted for investigating the water uptake in MFI zeolites, could be easily applied to investigate the transport of other guests' molecules into any nanoporous host system. In fact, while experiments (or simulations) alone cannot provide a full knowledge on the nature of surface and intracrystalline resistances,⁷ an integrated numerical/experimental approach allows clearly distinguishing and quantifying their respective effects.

Perspectives. The reported results unveil a promising future for zeolite-based membranes (see Supplementary Figure 11), because – by reducing surface diffusion resistances – they could potentially achieve orders of magnitude enhancements in membrane permeability respect to state-of-the-art polymeric membranes. To this purpose, in this work we highlight the key role of first understanding the chemical origin of surface barriers (whether they are an intrinsic property of synthesis or are caused by avoidable surface contamination), and then to introduce novel experimental techniques to reducing the hindered transport that occurs at the surface.

On the one hand, molecular-based simulations using zeolites as the membrane material typically utilize a 'perfect' (surface barrier free) crystal as the active membrane.^{25,26} Since these simulations have greatly exaggerated the performance of zeolites in comparison to the experiments, we believe that future simulations should incorporate this non-ideal structure within their model. However, a clear picture of surface structure of zeolites remains a challenge, because current imaging techniques are not yet precise enough to visualize experimentally such subnanometer surface terminations.^{8,9} Hence, computational quantum mechanics may be a helpful tool to determine the chemical characteristics of pore blockage or narrowing at the membrane surface, and it could be eventually coupled to molecular dynamics techniques to achieve a more accurate, multiscale simulation of surface diffusion resistances.

On the other hand, this work highlights the need for new surface treatments for zeolite-based membranes, which may reduce the surface pore blockage thus enhancing their permeability regardless the presence of intracrystalline defects. For example, faster molecular transport could be attained by post-synthetic processing, such as *ad hoc* surface etching;⁸ whereas clean synthesis and storage are also important to avoid the formation of surface barriers during samples' handling and operation.¹⁵

Q3.3 Although the results of the theoretical study in this work are good (a critical note on the model is the choice to represent acidic bridging hydroxyl groups as silanol defects; this is in my view not necessary [Al can be incorporated easily] and also incorrect as it overestimates the hydrophilicity!), they turn out to be not very relevant for the point to be made.

R3.3 Even though, from the chemical point of view, silanol nests and Al defects are different, this is a well-established practice in atomistic simulations.

In fact, in molecular dynamics partial charges of atoms are tunable input parameters, which have to be validated by either bottom-up (e.g., DFT calculations) or top-down (e.g., fitting of experimental results) approaches. Here, we decided (1) to keep the hydrophilic defect model as simple as possible, in order to capture the general effect of zeolite hydrophilicity on water adsorption, and (2) to tune the Si-OH

dipole moment (*i.e.*, zeolite hydrophilicity) by comparing simulated and experimental adsorption/infiltration isotherms of water in silicalite-I (Supplementary Figure 2).

Hence, instead of using different force field parameters for silanol nests (q_1) and Al (q_2) defects, it is a possible simulation practice to work with a single tunable parameter (*i.e.*, Si-OH dipole moment, in this case), to be tuned by global comparison with experimental data. In fact, the resulting effect (*i.e.*, zeolite hydrophilicity) to be compared with experiments would be $q = w_1 \times q_1 + w_2 \times q_2$, namely an infinite combination of the w_1 and w_2 weights. This is a simplified *top-down* approach to tune (and validate) molecular dynamics force fields to experimental data, namely an alternative to the *bottom-up* derivation from quantum mechanics simulations.

The silanol nest model was previously adopted in the literature to mimic MFI zeolites with increasing hydrophilicity, for example:

- Trzpit *et al.* ²⁷

“A molecular simulation study of a model silicalite-1 system, in which a silanol nest defect was introduced, enabled qualitative reproduction of the experimentally observed condensation thermodynamic features. [...]

*This silanol nest geometric structure is in very good qualitative agreement with the DFT calculations of Sokol *et al.*, as well as those of the previously published semiclassical simulations and *ab initio* cluster calculations.”*

- Cailliez *et al.* ²⁸

“In the present work, we investigate the defective silicalite-1 model in more detail, in order to gain insight into the effect of surface heterogeneity on the hydration behavior of a hydrophobic pore. The surface heterogeneity is tuned by introducing a variable amount of attractive (“hydrophilic”) defects, namely, silanol nests, in the model. [...]

It must be specified here that we are not considering the high defect content models as particularly realistic representations of true silicalite-1 materials. [...] Nevertheless, our aim here was to use these increasingly defective systems as toy models for examining the effect of increasingly heterogeneous inner surfaces on the water condensation thermodynamics.”

These (and further) clarifications and supporting references have been added in the methods sections.

REFERENCES

- 1 Sullivan, F. *Analysis of Global Desalination Market*, <<http://www.webcitation.org/6h7zwTs5m>> (2015).
- 2 Kärger, J. *et al.* Microimaging of transient guest profiles to monitor mass transfer in nanoporous materials. *Nat. Mater.* **13**, 333-343 (2014).
- 3 Kärger, J., Bülow, M., Millward, G. & Thomas, J. A phenomenological study of surface barriers in zeolites. *Zeolites* **6**, 146-150 (1986).
- 4 Hibbe, F. *et al.* The nature of surface barriers on nanoporous solids explored by microimaging of transient guest distributions. *J. Am. Chem. Soc.* **133**, 2804-2807 (2011).
- 5 Teixeira, A. R. *et al.* Dominance of Surface Barriers in Molecular Transport through Silicalite-1. *J. Phys. Chem. C* **117**, 25545-25555 (2013).
- 6 Karwacki, L. *et al.* Morphology-dependent zeolite intergrowth structures leading to distinct internal and outer-surface molecular diffusion barriers. *Nat. Mater.* **8**, 959-965 (2009).
- 7 Saint Remi, J. C. *et al.* The role of crystal diversity in understanding mass transfer in nanoporous materials. *Nat. Mater.* **15**, 401-406 (2016).
- 8 Teixeira, A. R. *et al.* 2D Surface Structures in Small Zeolite MFI Crystals. *Chem. Mater.* **27**, 4650-4660 (2015).
- 9 Teixeira, A. R. *et al.* On Asymmetric Surface Barriers in MFI Zeolites Revealed by Frequency Response. *J. Phys. Chem. C* **118**, 22166-22180 (2014).
- 10 Heinke, L. & Kärger, J. Correlating surface permeability with intracrystalline diffusivity in nanoporous solids. *Phys. Rev. Lett.* **106**, 074501 (2011).
- 11 Chmelik, C. *et al.* Ensemble measurement of diffusion: novel beauty and evidence. *ChemPhysChem* **10**, 2623-2627 (2009).
- 12 Heinke, L., Kortunov, P., Tzoulaki, D. & Kärger, J. Exchange dynamics at the interface of nanoporous materials with their surroundings. *Phys. Rev. Lett.* **99**, 228301 (2007).
- 13 Chmelik, C. *et al.* Mass transfer in a nanoscale material enhanced by an opposing flux. *Phys. Rev. Lett.* **104**, 085902 (2010).
- 14 Gueudré, L., Bats, N. & Jolimaître, E. Effect of surface resistance on cyclohexane uptake curves in Silicalite-1 crystals. *Microporous Mesoporous Mater.* **147**, 310-317 (2012).
- 15 Heinke, L., Gu, Z. & Wöll, C. The surface barrier phenomenon at the loading of metal-organic frameworks. *Nat. Commun.* **5**, 4562 (2014).
- 16 Tzoulaki, D., Schmidt, W., Wilczok, U. & Kärger, J. Formation of surface barriers on silicalite-1 crystal fragments by residual water vapour as probed with isobutane by interference microscopy. *Microporous Mesoporous Mater.* **110**, 72-76 (2008).
- 17 Chang, C.-C., Teixeira, A. R., Li, C., Dauenhauer, P. J. & Fan, W. Enhanced Molecular Transport in Hierarchical Silicalite-1. *Langmuir* **29**, 13943-13950 (2013).
- 18 Kärger, J. In-depth study of surface resistances in nanoporous materials by microscopic diffusion measurement. *Microporous Mesoporous Mater.* **189**, 126-135 (2013).
- 19 Schüring, A., Gulín-González, J., Vasenkov, S. & Fritzsche, S. Quantification of the mass-transfer coefficient of the external surface of zeolite crystals by molecular dynamics simulations and analytical treatment. *Microporous Mesoporous Mater.* **125**, 107-111 (2009).

- 20 Humplik, T., Raj, R., Maroo, S. C., Laoui, T. & Wang, E. N. Effect of Hydrophilic Defects on Water Transport in MFI Zeolites. *Langmuir* **30**, 6446-6453 (2014).
- 21 Ruthven, D. M. & Kaerger, J. Comment on “On Asymmetric Surface Barriers in MFI Zeolites Revealed by Frequency Response”. *J. Phys. Chem. C* **119**, 29201-29202 (2015).
- 22 Dauenhauer, P. J. Reply to “Comment on ‘On Asymmetric Surface Barriers in MFI Zeolites Revealed by Frequency Response’”. *J. Phys. Chem. C* **119**, 29203-29205 (2015).
- 23 Johan, E., Abadal, C. R., Sekulić, J., Chowdhury, S. R. & Blank, D. H. Transport mechanisms of water and organic solvents through microporous silica in the pervaporation of binary liquids. *Microporous Mesoporous Mater.* **65**, 197-208 (2003).
- 24 Humplik, T. *et al.* Nanostructured materials for water desalination. *Nanotechnology* **22**, 292001 (2011).
- 25 Turgman-Cohen, S., Araque, J. C., Hoek, E. M. & Escobedo, F. A. Molecular dynamics of equilibrium and pressure-driven transport properties of water through LTA-type zeolites. *Langmuir* **29**, 12389-12399 (2013).
- 26 Liu, Y. & Chen, X. High permeability and salt rejection reverse osmosis by a zeolite nanomembrane. *Phys. Chem. Chem. Phys.* **15**, 6817-6824 (2013).
- 27 Trzpit, M. *et al.* The Effect of Local Defects on Water Adsorption in Silicalite-1 Zeolite: A Joint Experimental and Molecular Simulation Study. *Langmuir* **23**, 10131-10139 (2007).
- 28 Cailliez, F., Stirnemann, G., Boutin, A., Demachy, I. & Fuchs, A. H. Does water condense in hydrophobic cavities? A molecular simulation study of hydration in heterogeneous nanopores. *J. Phys. Chem. C* **112**, 10435-10445 (2008).

Reviewers' Comments:

Reviewer #2 (Remarks to the Author)

I believe the authors addressed the issues raised at the previous review cycle (at least within the current stat-of-the-art understanding), and therefore I recommend to publish

Reviewer #3 (Remarks to the Author)

The authors have very convincingly updated the manuscript based on the referee comments. Although they did not take away all my initial criticism, I feel that they present a solid case in highlighting surface barriers which is indeed a very novel aspect of membrane usage.

I propose to accept this fine paper.

REVIEWER #2

Q2.1 I believe the authors addressed the issues raised at the previous review cycle (at least within the current state-of-the-art understanding), and therefore I recommend to publish.

R2.1 We thank the reviewer for these comments.

REVIEWER #3

Q3.1 The authors have very convincingly updated the manuscript based on the referee comments. Although they did not take away all my initial criticism, I feel that they present a solid case in highlighting surface barriers, which is indeed a very novel aspect of membrane usage. I propose to accept this fine paper.

R3.1 We are glad that the modifications implemented in the revised version of the manuscript have convinced the reviewer on both the relevance of the problem and the original approach adopted in this work. We thank the reviewer for having stimulated this fruitful discussion.